# Vaccine wastage rates and attributed factors in rural and urban areas in Uganda: Case of Mukono and Kalungu districts

Mackline Ninsiima[1,2]*, Michael Muhoozi[1], Henry Luzze[3], Simon Kasasa[1]

1 Department of Epidemiology and Biostatistics, School of Public Health, Makerere University, Kampala, Uganda, 2 Uganda Public Health Fellowship Program, Uganda National Institute of Public Health, Kampala, Uganda, 3 Ministry of Health, Kampala, Uganda

* nmackline@musph.ac.ug

## Abstract

Vaccine wastage remains a challenge to effective immunization; especially in low-income countries. We estimated vaccine wastage rates and assessed attributed factors in Mukono and Kalungu districts in Uganda. A mixed methods study design was utilised to estimate vaccine wastage rates for BGC, OPV, IPV, PCV, MR, DPT–HepB–Hib for 6 months prospectively from March–August 2022 and assess attributed factors in 22 health facilities. Mann–Kendall statistical test was used to assess significance of observed trends. We applied Mann Whitney U and Kruskal–Wallis H tests to compare vaccine wastage rates per vaccine by district, ownership, and type of health facility. Additionally, we administered a questionnaire among 57 health workers and conducted 15 Key Informant Interviews to understand reasons for vaccine wastage. Overall vaccine wastage rates were BCG (70%), MR (58%), DPT–HepB–Hib (21%), IPV (31%), OPV (28%) and PCV (17%); exceeding accepted vaccine wastage rates in Kalungu and Mukono districts. Significant variations were observed across the different types of health facilities [BCG (p < 0.001), IPV (p = 0.023), MR (p = 0.004) and OPV (0.008)] and among health facilities located within urban and rural areas [BCG (p < 0.001), MR (p < 0.001) and OPV (0.003)]. Vaccine wastage rates for BCG and MR vaccines were higher compared to other vaccines because remaining doses in opened vials were discarded within 6 hours of reconstitution, as per the Multi Dose Vial Policy (MDVP). Other contributing factors were low turn up during outreaches, errors and non–completion of vaccine monitoring tools, cold chain failures and inadequate training in vaccine management. Vaccine wastage rates for all vaccines were relatively higher than acceptable levels in both districts. Intensified efforts such as regular review of vial opening guidelines, predictive modelling for outreach planning, decentralized vaccination approaches, and availability of vaccines in reduced-volume multi-dose vials where feasible could minimize vaccine wastage.

**Data availability statement:** All relevant data are within the manuscript and its Supporting Information files.

**Funding:** This project was supported by East African Regional Centre of Excellence for Vaccines, Immunization, Health Supply Chain Management under the Cooperative Agreement number MUSPH# 1025 through Makerere University School of Public Health. The funds were received by Dr. Simon Kasasa, the Principal Investigator for the project through the accounts department, Makerere University School of Public Health. Its contents are solely the responsibility of authors and do not necessarily represent the official views of East African Regional Centre of Excellence for Vaccines, Immunization, Health Supply Chain Management, Makerere University School of Public Health, or Ministry of Health. The staff of the funding body provided technical guidance in design of the study, ethical clearance, data collection, analysis, interpretation of data, and writing the manuscript.

**Competing interests:** The authors have declared that no competing interests exist.

## Background

Vaccination services are delivered by skilled healthcare professionals, whose numbers vary based on the level of care provided, both in public and private healthcare facilities. The vaccination schedule aligns with WHO guidelines to ensure comprehensive disease coverage. Immunization starts at birth with BCG and OPV vaccines. At 6, 10, and 14 weeks, children receive OPV, IPV, Rotavirus, DTP-HepB-Hib, and PCV vaccines. At 9 and 18 months, they receive the Measles-Rubella vaccine. Most vaccines, including BCG, OPV, IPV, Rotavirus, DTP-HepB-Hib, and Measles-Rubella, are supplied in multi-dose vials, while PCV is administered using single-dose vials [1]. In 2019, Uganda's vaccination coverage estimates indicated commendable performance, particularly with third dose Diphtheria, Tetanus, and Pertussis (DPT3) at 93%, Pneumococcal Conjugate vaccine (PCV3) at 92%, Polio 3 at 92%, BCG at 88%, completed dose of Rotavirus vaccines at 87%, and first dose of Measles-containing-vaccine (MCV1) at 87% [2]. During the COVID-19 pandemic, there was a decline in routine immunization coverage [3].

Vaccine wastage, defined as the proportion of doses discarded in opened or unopened vaccine vials that are not used to vaccinate an eligible individual, contributes to missed opportunities to vaccinate (MOV) [4]. Vaccine wastage has significant financial consequences, impacting healthcare budgets and potentially hindering immunization program sustainability. Vaccine wastage significantly impacts the cost-effectiveness of vaccine procurement, as each dose wasted represents a loss of the initial investment in vaccine procurement, necessitating additional procurement to compensate for wasted doses. Each wasted dose not only represents a loss in terms of the vaccine itself but also the resources invested in its handling and delivery, such as storage, transportation, and distribution. The financial burden of managing and disposing of wasted vaccines further impacts on the already constrained health budgets by diverting resources away from other essential health services.

General guidelines for Vaccine Wastage Rates (VWR) recommend acceptable rates for different vaccines: 50% for BCG and 25% for reconstituted measles vaccine, 10% for OPV, 15% for liquid vaccines in multi-dose vials containing 10 or more doses, and 5% for liquid vaccines in single or two-dose vials like PCV [5,6]. Balancing the efficient provision of vaccines to targeted children with minimizing wastage from multi-dose vials is challenging because while reducing wastage is desirable, the primary goal is to ensure high vaccination coverage. Some level of wastage is acceptable and necessary to achieve broader public health objectives, as prioritizing coverage over wastage helps protect more children from preventable diseases. The Ministry of Health recommends that health workers open a multi dose vial even if only one child turns up at the immunization centre to reduce missed opportunities for vaccination. While the aim is to optimize vaccine coverage, the safety requirement to dispose opened multi-dose vials at the end of a vaccination session or within six hours for vaccines without preservatives, unless the vaccine meets the criteria for use for up to 28 days after opening, leads to significant vaccine wastage rates [7]. Vaccine usage and wastage should be monitored monthly at all service points. Following the development of tools for estimating vaccine

wastage by UNEPI, only 11% of health facilities calculated wastage rates, largely due to the absence of decentralized training sessions aimed at instructing healthcare workers on the methodology for calculating vaccine wastage [8]. To this effect, Uganda does not have locally generated vaccine wastage estimates that can guide forecasting and need estimation. Consequently, the country has continued to rely on WHO projected wastage rates to estimate vaccine needs.

Under estimation of vaccines can lead to stock outs of a particular vaccine and increase missed opportunities for vaccination (MOV), one of the contributing factors to low vaccination coverage and eventual vaccine preventable disease outbreaks. On the other hand, over estimation of vaccines can lead to over stocking of a particular vaccine, leading to vaccine expiry before utilization. Estimating vaccine wastage rates for different vaccines can guide planners in accurately estimating the vaccine needs, hence maximizing utilization of available resources [9]. Of note, health workers are expected to ensure that vaccine wastages are minimized to acceptable levels [10]. This underscores the urgent need to estimate vaccine wastage rates and understand attributable factors among health workers to inform policy efforts towards minimising vaccine wastage in Uganda.

Additionally, existing literature indicates that logistical challenges, infrastructure limitations, and variations in healthcare delivery systems between rural and urban areas could influence vaccine wastage rates [11,12]. In low- and middle-income countries, vaccine wastage rates were significantly higher in rural areas compared to urban areas [5,13–17]. This discrepancy was attributed to inadequate cold chain infrastructure, health care access, longer travel distances, population density, vaccine delivery logistics and limited healthcare facilities in rural settings, all of which could contribute to increased wastage. Furthermore, variations in population density between rural and urban areas can impact the efficiency of vaccine distribution and utilization. Based on these findings, we hypothesized that vaccine wastage rates would be higher in rural settings compared to urban settings in Uganda. This informed the decision to select Mukono and Kalungu districts to represent urban and rural settings respectively. We estimated vaccine wastage rates and assessed attributed factors among health workers participating in vaccination activities in Mukono and Kalungu districts in Uganda.

## Methods

### Study site

The assessment was carried out in randomly sampled health facilities providing vaccination services in Mukono and Kalungu; the urban and rural districts respectively. Data collected from the two districts also provided an opportunity to compare vaccine wastage rates to compare potential disparities with respect to rural and urban, type of health facility (hospitals and health centres classified as II, III, and IV) and ownership (private versus public health facilities) variations in Uganda. Mukono and Kalungu districts were grouped under Category 1 with good vaccine access and vaccine utilization as per the 2019/20 WHO Reaching Every District (RED) Categorization for measuring routine vaccination performance [18]. Selection from the same category counteracted any inherent differences which could have been attributed to performance. The two districts are located in the central part of the country. Mukono district has a higher number of health facilities, forty–five compared to Kalungu with twenty–two health facilities providing vaccination services. According to the 2014 Population and Housing census, the population of Mukono was almost double that of Kalungu [19].

### Study design and sampling procedures

Mixed methods study design was utilised to estimate vaccine wastage rates and assessed attributed factors among health workers participating in vaccination activities in Mukono and Kalungu districts in Uganda. The study team took additional measures to control for the data quality since routine vaccine data suffers incompleteness and underreporting due to concerns about perceived performance indicators. The study team therefore directly engaged with health workers, did periodic follow-up, and targeted support to ensure completeness and accuracy of recorded vaccine wastage data. Unlike routine vaccine monitoring, which primarily relies on passive reporting, this study adopted a more structured approach by integrating periodic facility engagement, reinforcement of SOPs, and proactive data verification.

### Quantitative phase

We computed monthly vaccine wastage rates for BGC, OPV, IPV, PCV, MR, and DPT–HepB–Hib for a period of 6 months. Furthermore, we compared vaccine wastage rates by type of health facility, ownership and district, and assessed attributed factors among health workers participating in vaccination activities. For purposes of triangulation, the qualitative phase explored reasons for vaccine wastage among health workers participating in vaccination activities in the two districts.

At the district level, a roster of health facilities categorized by their respective levels (including hospitals and health centres designated as II, III, and IV) was compiled using data extracted from the DHIS2 database. Health facilities that had not been providing vaccination services for the past six months were excluded. The decision to collect data for six months in our longitudinal study in only two districts was based on the need to balance the objectives of observing trends over time with the practical considerations of financial constraints. The investigative team believed that six months would be a sufficient reasonable time frame to observe, analyse trends and detect any noticeable patterns or fluctuations in wastage rates. While a longer duration might have provided more comprehensive insights, we aimed to maximize the utility of available resources while still striving to achieve the study's objectives effectively.

Based on the DHIS2 database, Mukono and Kalungu districts had forty–five and twenty–two health facilities respectively providing vaccination services as of December 2022. To ensure a representative sample, we used proportionate to size sampling to help generate an initial pool of health facilities where random sample was to be drawn. Using a table of random number generator, a third of eligible public and private health facilities were randomly selected from the list of health facilities per district. Fifteen health facilities, comprising 3 hospitals, 7 HC IIIs, 1 HC IV, and 4 HC IIs, were randomly chosen from Mukono district. Similarly, seven health facilities, including 1 hospital, 1 HC IV, 3 HC IIIs, and 2 HC IIs, were randomly selected from Kalungu district, all of which offered vaccination services. This approach ensured that the sample size was proportionate to the total number of facilities in each district, providing a balanced representation of different facility types. Additionally, we administered a questionnaire among 57 health workers providing vaccination services to identify reasons for vaccine wastage in selected health facilities.

### Qualitative phase

Eligible participants for the Key Informant Interviews included two Assistant District Health Officers in charge of maternal and child health services at the district level and twenty-two immunization focal persons from selected health facilities. Out of twenty-four eligible key informants, twelve Key Informant Interviews were conducted based on the data saturation point; a point at which further sampling did not generate any new concepts or ideas about the phenomenon under investigation.

## Data collection

### Quantitative phase

Vaccine monitoring data for BGC, OPV, IPV, PCV, MR and DPT–HepB–Hib were extracted from Vaccine and Material Injection Control Book (VIMCB) and tally sheets using the abstraction tool from March–August 2022. Months abstracted represented both a rainy season (March–May) and a dry season (June–August). Uganda typically experiences two rainy seasons: March–May and September–November. Given that rainfall has been associated with lower vaccination turnout, the inclusion of both rainy and dry months in the study period allowed for an assessment of potential seasonal fluctuations in vaccine wastage rates. Information collected on a monthly basis required to calculate the monthly vaccine wastage rate as per the WHO recommended formula included: number of usable doses at the beginning of the month/starting monthly balance of vaccine doses, number of vaccine doses received during the month, number of usable doses in stock at the end of the month/monthly closing balance of doses, and number of doses administered/number of children vaccinated. A questionnaire, developed based on available literature, was administered once at the beginning of the study

by experienced research officers among 57 health workers providing vaccination services in the selected health facilities to identify reasons for vaccine wastage. The study team engaged health workers at the selected facilities in structured orientation sessions before the data collection period to ensure consistent documentation of vaccine usage and wastage. These sessions were based on standard operating procedures for vaccine monitoring which emphasized the correct use of the VIMCB and tally sheets. The research team worked closely with facility in charges to facilitate adherence to best practices in vaccine documentation. During the study period, periodic reinforcement was provided through monthly sched-uled supervisory visits by the research team. Facility-based supervisors (Health Facility Expanded Programme on Vac-cination (EPI) focal personnel) were encouraged to remind health workers of proper vaccine documentation procedures, particularly during weekly staff meetings using telephone calls by research team. check-ins phone calls were also used to address any emerging challenges in vaccine usage and wastage reporting. Extracted vaccine usage and wastage records were cross-validated with stock registers and verified against administrative records at the district health stores with goods received and delivery notes from National Medical Store of Uganda. Where discrepancies arose, follow-up discussions were conducted with facility in-charges to ascertain accurate records.

## Qualitative phase

Qualitative data was collected in a single session at the beginning of the study using the key informant interview guide among Assistant District Health Officers and immunization focal persons at the selected health facilities. The Key Infor-mant Interview guide was specifically developed for this study with questions aimed at eliciting reasons for vaccine wast-age in health facilities. The Key Informant Interview guide was pre–tested and recommended changes in the questions were made to ensure that the guide captured relevant and appropriate information before use. Key Informant Interviews were conducted in English and audio–recorded on receipt of informed consent. Verbatim transcripts were proof–read before importing them into Atlas.ti version 6.0, a qualitative data management software.

## Data analysis

The monthly vaccine wastage rate was computed based on WHO–recommended formula for calculating wastage rate [4].

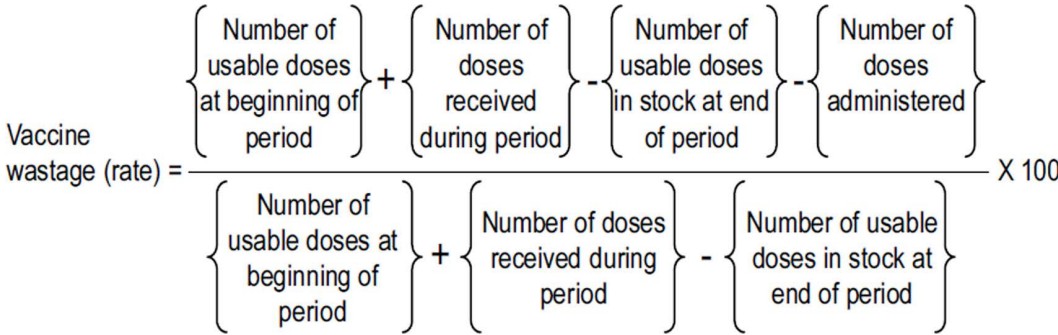

$$\text{Vaccine wastage (rate)} = \frac{\left\{\begin{array}{c}\text{Number of} \\ \text{usable doses} \\ \text{at beginning of} \\ \text{period}\end{array}\right\} + \left\{\begin{array}{c}\text{Number of} \\ \text{doses} \\ \text{received} \\ \text{during period}\end{array}\right\} - \left\{\begin{array}{c}\text{Number of} \\ \text{usable doses} \\ \text{in stock at end} \\ \text{of period}\end{array}\right\} - \left\{\begin{array}{c}\text{Number of} \\ \text{doses} \\ \text{administered}\end{array}\right\}}{\left\{\begin{array}{c}\text{Number of} \\ \text{usable doses at} \\ \text{beginning of} \\ \text{period}\end{array}\right\} + \left\{\begin{array}{c}\text{Number of doses} \\ \text{received during} \\ \text{period}\end{array}\right\} - \left\{\begin{array}{c}\text{Number of usable} \\ \text{doses in stock at} \\ \text{end of period}\end{array}\right\}} \times 100$$

At the health facility level, monthly vaccine wastage rates for BGC, OPV, IPV, PCV, MR and DPT–HepB–Hib were com-puted from March–August 2022. At the district level, average vaccine wastage rate for each vaccine per month was calcu-lated by adding vaccine wastage rates from selected health facilities divided by the number of health facilities. The overall average vaccine wastage rate for each vaccine was determined by adding average vaccine wastage rates per month divided by the number of months. Mann–Kendall statistical test was utilized to assess the significance of observed trends of vaccine wastage rates based on Kendall's tau correlation coefficient (r) and p–values. We applied Mann Whitney U test to compare overall average vaccine wastage rates for BGC, OPV, IPV, PCV, MR and DPT–HepB–Hib by type of district (rural versus urban districts) and ownership (private versus public health facilities). For type of health facility, we applied

Kruskal–Wallis H test to compare vaccine wastage rates since we had more than two categories of health facilities. This statistical test was selected to determine any differences in the average vaccine wastage rates because outcome data was not normally distributed. Significance was determined at a p–value of <0.05. Furthermore, we calculated proportions of reasons for both closed– and open–vial vaccine wastage reported by health workers. Data analysis was conducted using Stata/SE Version 14.0.

Thematic analysis method using inductive coding was used to analyse qualitative data. Exploration of data and synthesis of codes, subthemes and themes was done in Atlas.ti Version 6.0. Relevant verbatim quotations were selected as evidence to support the generated themes.

### Ethical approval and consent to participate

Approval to conduct the study was obtained from the Makerere University School of Public Health Higher Degrees Research and Ethics Committee (HDREC). Through the Uganda National Expanded Program on Immunization (UNEPI), Ministry of Health, administrative clearance was sought from the offices of the District Health Officer (DHO) and Health Facility in–charges before conducting data collection. The research team ensured that all eligible participants provided written informed consent, and all interviews were conducted on a voluntary basis. A copy of signed informed consent was given to each respondent.

Interviewees were given the option to refuse to respond to any questions or the whole interview, if at any time they believed a response would contain sensitive information or if they were not comfortable to answer. Responses were not linked to any specific person in the final report and all information provided was kept confidential and used for planning purposes only. The investigative team did not share responses of specific individuals with their co–workers and they were assured that whatever was said would not jeopardize their employment. Information that could be directly linked to individuals was not documented in this manuscript.

## Results

### Vaccine wastage rates

The overall average vaccine wastage rates were: BCG (70%), DPT–HepB–Hib (21%), IPV (31%), MR (58%), OPV (28%) and PCV (17%) in Kalungu and Mukono districts from March–August, 2022. Overall vaccine wastage rate for BCG (70%) was the highest, followed by Measles Rubella (58%) among all vaccines (Table 1). Notably, average vaccine wastage rates in Kalungu and Mukono districts exceeded the acceptable rates during the evaluation period.

### Monthly average vaccine wastage rates, March–August 2022

In Kalungu district, average vaccine wastage rates for BCG and Measles Rubella vaccines were higher compared to other vaccines from March–August 2022 (Fig 1). The IPV vaccine demonstrated a significant increase in wastage in

**Table 1. Vaccine wastage rates in Kalungu and Mukono districts, March–August 2022.**

| Vaccine | Acceptable vaccine wastage rates | Vaccine wastage rates (%) | | | | Overall average vaccine wastage rates (%) |
|---|---|---|---|---|---|---|
| | | Kalungu | | Mukono | | |
| | | Range | Average | Range | Average | |
| BCG | 50 | 67 – 82 | 77 | 60 – 67 | 63 | 70 |
| DPT–HepB–Hib | 15 | 10 – 43 | 23 | 10 – 27 | 19 | 21 |
| IPV | 15 | 20 – 68 | 33 | 17 – 38 | 28 | 31 |
| MR | 25 | 61 – 73 | 66 | 41 – 62 | 50 | 58 |
| OPV | 10 | 20 – 41 | 33 | 18 – 30 | 22 | 28 |
| PCV | 5 | 12 – 37 | 19 | 9 – 24 | 15 | 17 |

June followed by a decline in July, while the BCG vaccine exhibited its highest wastage rate in August compared to other months and vaccines. In July, there seems to be a noticeable decrease in the wastage rates for most vaccines compared to the previous months.

In Mukono district, average vaccine wastage rates for BCG and Measles Rubella vaccines were higher compared to other vaccines from March–August 2022 (Fig 2). Over the months, there are variations in wastage rates for each vaccine, which do not follow a uniform trend across all vaccines.

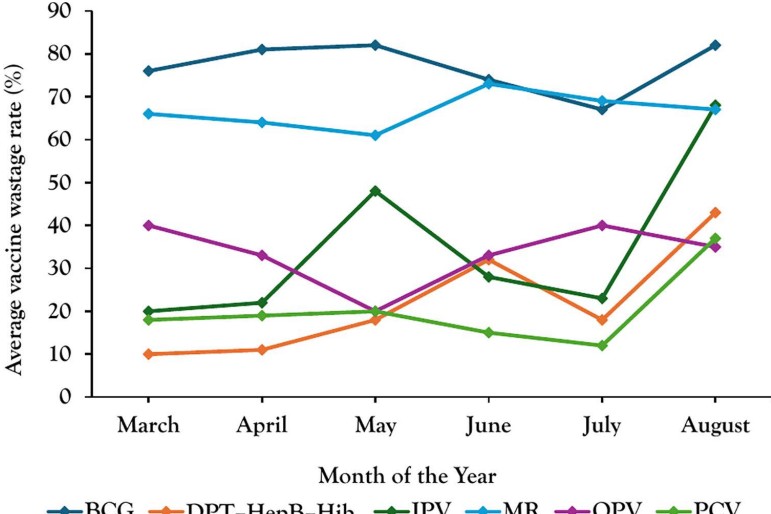

**Fig 1. Monthly average vaccine wastage rates in Kalungu District, March–August 2022.**

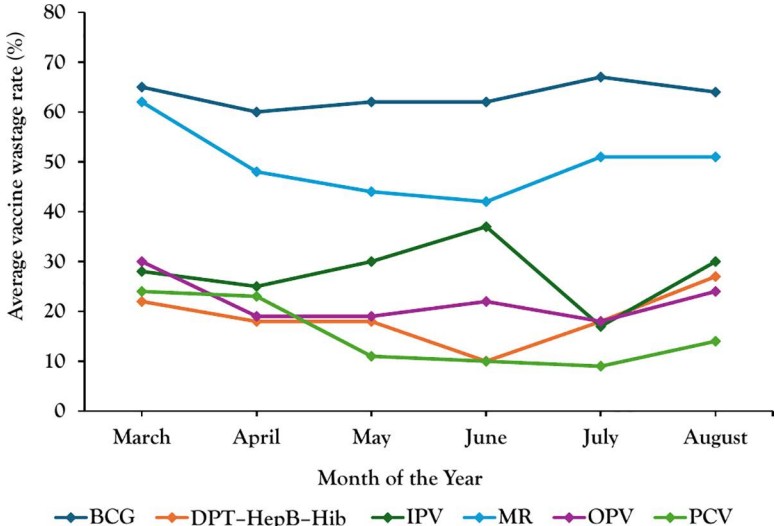

**Fig 2. Monthly average vaccine wastage rates in Mukono District, March–August 2022.**

**Trends of vaccine wastage, March–August 2022**

Despite increasing and decreasing trends observed based on the Mann–Kendall statistic and Kendall's tau correlation coefficient, they were not statistically significant for all vaccines in Kalungu and Mukono districts ([Table 2]).

**Comparison of average vaccine wastage rates by district and ownership, March–August 2022**

Vaccine wastage rates for all vaccines were higher in Kalungu District compared to Mukono District. Significant differences in average vaccine wastage rates for BCG, MR and OPV were observed between Kalungu and Mukono districts ([Table 3]). There were no statistically significant differences in average vaccine wastage rates for all vaccines between public and private health facilities.

**Table 2. Trends of vaccine wastage, March–August 2022.**

| Vaccine | Mann–Kendall statistic (S) | Kendall's tau correlation coefficient | p–value |
|---|---|---|---|
| **Kalungu District** | | | |
| BCG | 276 | 0.09 | 0.235 |
| DPT–HepB–Hib | 35 | 0.05 | 0.692 |
| IPV | 46 | 0.04 | 0.680 |
| MR | −122 | −0.05 | 0.548 |
| OPV | −41 | −0.02 | 0.821 |
| PCV | −120 | −0.15 | 0.181 |
| **Mukono District** | | | |
| BCG | 28 | 0.04 | 0.734 |
| DPT–HepB–Hib | 45 | 0.09 | 0.495 |
| IPV | −6 | −0.01 | 0.932 |
| MR | 136 | 0.18 | 0.102 |
| OPV | 17 | 0.03 | 0.834 |
| PCV | −24 | −0.09 | 0.568 |

**Table 3. Comparison of average vaccine wastage rates by district and ownership, March–August 2022.**

| | Vaccine wastage rate (%) | | p–value |
|---|---|---|---|
| **District** | **Kalungu (Rural)** | **Mukono (Urban)** | |
| BCG | 77 | 63 | **<0.001*** |
| DPT–HepB–Hib | 23 | 19 | 0.688 |
| IPV | 33 | 28 | 0.719 |
| MR | 66 | 50 | **<0.001*** |
| OPV | 33 | 22 | **0.003** |
| PCV | 19 | 15 | 0.551 |
| **Ownership** | **Public HFs** | **Private HFs** | |
| BCG | 72 | 67 | 0.845 |
| DPT–HepB–Hib | 13 | 23 | 0.051 |
| IPV | 29 | 30 | 0.087 |
| MR | 55 | 56 | 0.681 |
| OPV | 22 | 28 | 0.188 |
| PCV | 14 | 17 | 0.138 |

*p < 0.05 ** p < 0.01 *** p < 0.001

## Comparison of average vaccine wastage rates by type of health facility, March–August 2022

Significant differences in average vaccine wastage rates for BCG, IPV, MR and OPV were observed between different types of health facilities (Table 4). Vaccine wastage rates were higher in level II health centres across all vaccines.

## Characteristics of survey participants

Out of 57 health workers, most health workers (n = 30, 52.6%) were providing vaccination services in Health Centre IIIs. Of note, (n = 39, 68.4%) had only completed certificate programs whereas 47.4% served as enrolled midwives or nurses. Based on self–reports, (n = 20, 35.1%) of the respondents had participated in vaccination activities for 10 years and above.

## Reasons for vaccine wastage

The reasons for vaccine wastage presented in this section were obtained from self-reported responses collected through questionnaires administered to health workers providing vaccination services in the selected health facilities. For closed vial vaccine wastage, the primary contributing factors identified were expiry of vaccines (14%), vaccine storage space constraints (11%), and cold chain failure (2%) (Fig 3). On the contrary, open vial vaccine wastage exhibited a more varied set of reasons, with discarding reconstituted vials after 6 hours and discarding open vials after vaccination being the most prevalent factors, at 96% and 93% respectively.

**Table 4. Comparison of average vaccine wastage rates by type of health facility, March–August 2022.**

| Vaccine | Vaccine wastage rate (%) | | | | p–value |
|---|---|---|---|---|---|
| | Hospital | HC IV | HC III | HC II | |
| BCG | 51 | 66 | 68 | 82 | **<0.001\*\*\*** |
| DPT–HepB–Hib | 16 | 36 | 17 | 21 | 0.488 |
| IPV | 17 | 27 | 25 | 52 | **0.023\*** |
| MR | 50 | 66 | 51 | 63 | **0.004\*\*** |
| OPV | 20 | 41 | 23 | 26 | **0.008\*\*** |
| PCV | 12 | 23 | 15 | 18 | 0.235 |

*\*p < 0.05 \*\* p < 0.01 \*\*\* p < 0.001*

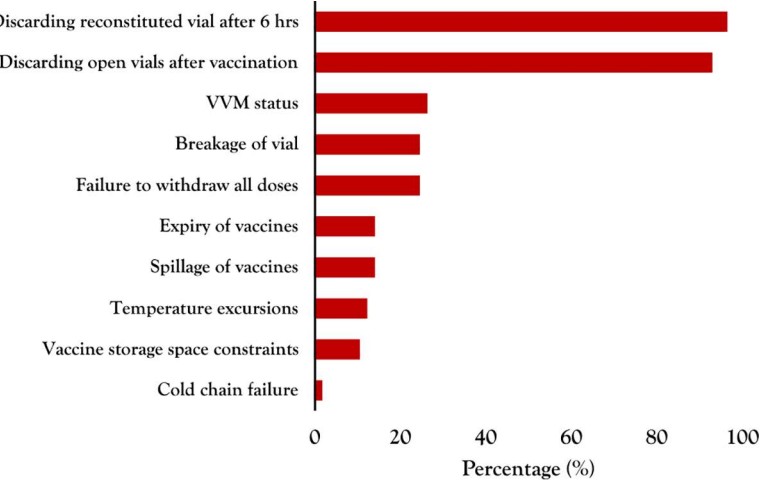

**Fig 3. Reasons for vaccine wastage in Mukono and Kalungu Districts.**

## Characteristics of key informants

Most key informants were Assistant Nursing Officers (n = 7, 47%) whereas the majority (n = 9, 60%) had provided vaccination services for 5 – 9 years (Table 5).

Reasons for vaccine wastage highlighted during Key Informant Interviews among immunization focal persons in selected health facilities in Kalungu and Mukono Districts have been stipulated below:

### High vaccine dosage vial

All immunization focal persons attributed vaccine wastage to high vaccine vial dosage coupled with the practice of opening a vial even if there is one child at the vaccination session to reduce missed opportunities among eligible children. BCG and Measles Rubella vaccines were the most affected vaccines since their opened multi–dose vials should be discarded at the end of the immunization session, or within six hours of opening.

*"…they always say that even if you get one kid please endeavor to immunize, when you open up that vaccine you immunize one or two kids the rest of the vaccine is wasted, that is where I see wastage."* **KII 06, Enrolled Midwife**

*"If a mother delivers a baby, that baby must be vaccinated before the mother is discharged. Yes, because we don't want this child to miss BCG vaccine. But do you what this means… It means that after vaccinating this baby, I will waste the remaining 9 doses because I have to discard them after 6 hours…"* **KII 07, Assistant Nursing Officer**

### Low turnup during vaccination outreaches

Most immunization focal persons reported that they had encountered wastages during vaccination outreaches attributed to low turnup of eligible children. Despite continued efforts to utilize Village Health Teams (VHTs) for mobilization, health workers do not meet the targeted numbers of children expected increasing the chances for vaccine wastage at the end of vaccination outreaches.

*"Yes, we involve them (VHTs) but the main issue during rainy season we don't get much we don't get many children because the parents are always in the gardens, yeah… their turn up is low. We don't get many… sometimes, you have opened a Measles Rubella vaccine for one child… And you are forced to discard the remaining doses because other eligible children did not turn up."* **KII 03, Assistant Nursing Officer**

Table 5. Characteristics of immunization focal persons who participated in Key informant interviews in Kalungu and Mukono districts.

| Variable | Frequency (n = 15) | Percentage (%) |
|---|---|---|
| **Cadre of health worker** | | |
| Assistant District Health Officer (MCH) | 2 | 13 |
| Assistant Nursing Officer | 7 | 47 |
| Enrolled Midwife | 3 | 20 |
| Nursing Assistant | 3 | 20 |
| **Years of service in vaccination activities** | | |
| 2 – 4 | 1 | 7 |
| 5 – 9 | 9 | 60 |
| 10 and above | 5 | 33 |

### Errors and non–completion of vaccine monitoring tools

The majority of immunization focal persons reported non–completion and inconsistencies in documenting receipt and issuing of vaccines hence underestimation or overestimation of vaccine wastage rates. Even though health workers providing vaccination activities have been trained in completing vaccine monitoring tools, vaccines received and those issued out have not been accurately recorded in Vaccine and Material Injection Control Book. Accurately estimating vaccine wastage rates in these facilities is difficult due to poor data generation and quality.

*"… actually, look at the tally sheets and the control book they speak totally a different language so you might record it as a wastage but in actual sense it is what was given out but it was not recorded properly…"* **KII 15, Assistant District Health Officer (MCH**)

### Failure in maintaining cold chain

Some immunization focal persons complained of frequent power fluctuations which destabilize the cold chain system. Such scenarios have led to wastage of batches of vaccines especially in lower health facilities without alternative sources of power like generators and solar fridges.

*"In our facility, we have not yet received a solar fridge… And yet, sometimes we don't have electricity. Recently, there was no electricity throughout the day till the next day… We were forced to discard the vaccines because of the undesirable red signal off the Vaccine Vial Monitor"* **KII 11, Assistant Nursing Officer**

In such instances, health workers revealed that they had to transfer vaccines to health facilities with power or any other places of convenience. One of the key informants said,

*"…for sometimes power goes off and we don't have back up but we keep on monitoring our what, we keep on monitoring our fridge when we see the temperature is going up, we take our vaccines to another health facility with power… I remember there was a time we had to transfer fridges to the police station so that we maintain the cold chain."* **KII 03, Assistant Nursing Officer**

### Lack of training in vaccine management

Several immunization focal persons reported that most of the health workers providing immunization services have not received training in vaccine management with an objective of minimizing vaccine wastage. Of note, some of the health workers who previously received the training have been transferred to other units whereas others were terminated or resigned especially in private health facilities.

*"You see… most of us have never been trained on vaccine wastage issues… Whatever little we know, we had to ask our colleagues in the same health facility or even call in another health facility… Sometimes we are not even sure whether what we are doing is the correct thing to do…"* **KII 04, Assistant Nursing Officer**

One of the district officials commented on poor reconstitution practices by health workers who are new and inexperienced saying that,

*"…of late we have gotten more of young nurses coming on board and the expertise of giving a shot is another issue so they could be drawing more, …. so the vaccination skill is also another reason which contributes to the wastage as well…"* **KII 01, Assistant District Health Officer (MCH)**

## Discussion

Overall vaccine wastage rates exceeded accepted vaccine wastage rates in Kalungu and Mukono districts from March–August 2022. Significant variations in vaccine wastage rates were observed across the different types of health facilities and further, among health facilities located within urban and rural areas. Vaccine wastage rate for BCG was the highest, followed by Measles Rubella compared to DPT–HepB–Hib, IPV, OPV and PCV. Vaccine wastage rates particularly for BCG and MR vaccines were higher compared to other vaccines because remaining doses in opened vials of these reconstituted vaccines without preservatives have to be discarded within 6 hours of reconstitution, as per the Multi Dose Vial Policy (MDVP). [7]. Other contributing factors were low turn up during vaccination outreaches; failure in maintaining cold chain and lack of training in vaccine management among health workers.

Acceptable vaccine wastage rates are: 50% for BCG; 15% for DPT–HepB–Hib and IPV; 5% for PCV; 25% for Measles vaccine; and 10% for OPV [5,6]. In this evaluation, overall vaccine wastage rates were higher than acceptable vaccine wastage rates, contrary to findings from Cameroon where wastage estimates during 2016 and 2017 were at an acceptable level [5].. It's quite difficult to compare estimated vaccine wastage rates with general projected wastage rates and mathematical modelling to ascertain the discrepancies in Uganda. Country specific vaccine wastage rates have not been provided since data regarding vaccine wastage rates has been sporadic and unreliable. Such evidence gaps make it difficult to reliably forecast and redistribute antigens since there are no realistic estimates of vaccine wastage rates to guide procurement and supply of vaccine.

High vaccine wastage rates for BCG and MR compared to other vaccines have been reported; demonstrating coherence to existing literature from Cameroon, Gambia and Bangladesh [5,20]. Multi–dose vaccine vials exhibit higher vaccine wastage rates compared to single or lower dose vaccine vials unless they are used in mass vaccination activities. Transition from 5– to 10–doses vials for rotavirus vaccine increased vaccine wastage rates in India [21]. Very slight differences between estimated and permissible vaccine wastage rates for BCG and MR were attributed to smaller 10 and 5–dose vials. However, there were large differences between IPV and OPV vaccine wastage rates and acceptable levels due to high 25–dose vaccine vial presentation [22]. In the Ugandan immunization setting, OPV is frequently used in routine immunization and supplementary immunization activities, leading to higher utilization and potentially lower wastage compared to IPV. IPV, on the other hand, is administered less frequently and in smaller target populations, meaning that open vials are less likely to be fully utilized before expiry, contributing to higher wastage rates. Additionally, unlike OPV, which is administered orally and can be used across different vaccination sessions, IPV requires intramuscular administration, making dose usage more constrained in settings with lower patient volumes. Evidence based findings indicate that high number of doses in a vaccine vial increase the chances for vaccine wastage, unless the vaccine meets the criteria for use for up to 28 days after opening [23].

Health workers proposed several strategies to balance the efficient provision of vaccines to targeted children with the need to minimize vaccine wastage, particularly for multi-dose vaccines. Improve planning for vaccination outreaches by employing predictive modelling and data analytics to estimate the number of attendees accurately. This can help health workers anticipate demand, optimize resource allocation, and minimize the risk of vaccine wastage at the end of the session. Explore the feasibility of setting up mobile vaccination clinics or pop-up sites in areas with lower turnouts. This decentralized approach, organized in collaboration with communities, can bring immunization services closer to local populations, potentially increasing the number of children reached and minimizing wastage associated with centralized sessions.

Public health authorities should advocate for the manufacture of vaccines in reduced-volume multi-dose vials where feasible [9,24]. While this might not be applicable to all vaccines, it can be a viable solution eliminating the challenge associated with high volume multi-dose vials. Establish a mechanism for regular review of immunization guidelines to ensure they align with current realities and technological advancements. This includes revisiting policies on vial openings, storage, and disposal to adapt to the evolving needs of the immunization program. These strategies should be prioritized

to ensure successful roll out of upcoming multi dose vaccines, ensuring the benefits of vaccination reach as many individuals as possible while minimizing wastage. Our finding of higher vaccine wastage in a rural district compared to the urban districts is consistent with earlier findings. Peer reviewed studies and reports have reported variations across vaccine wastage rates among health facilities located within the urban and rural areas [25–27]. Vaccine wastage rates are higher in rural areas compared to urban areas. These discrepancies could be explained by population density, coverage and frequencies of providing vaccination. Sparse populations and higher number of vaccination outreaches to complement static vaccination days have been highlighted among contributing factors to vaccine wastage in rural areas [5,17,21]. Furthermore, differences in vaccine wastage rates have also been observed across the different types of health facilities. What is quite challenging was that some health facilities share targeted coverage areas and conduct vaccination outreaches in the same places; hence increasing incidences for wastages since expected eligible children might have been vaccinated by health workers from another health facility. In this regard, health workers should update micro plans to cater for recent changes in target populations and avoid overestimation of expected number of eligible children. Health workers should also engage in collaborative efforts with their counterparts in neighboring facilities to streamline vaccine service delivery and ensure comprehensive coverage within overlapping areas during the microplanning process.

The observed significant differences in average vaccine wastage rates among different types of health facilities, particularly higher rates in level II health centres across all vaccines, suggest potential disparities in vaccine management practices and resource utilization. Several factors may contribute to the higher wastage rates in level II health centres. These facilities often cater to smaller populations and may face challenges in maintaining optimal stock levels due to fluctuating demand. Additionally, limited storage capacity and inadequate infrastructure for cold chain management in level II health centres may contribute to vaccine spoilage and wastage. Furthermore, differences in vaccine delivery practices, such as handling procedures, storage conditions, and staff training, may influence wastage rates across health facility types. Level II health centres, with potentially fewer resources and staff compared to higher-level facilities, may face greater challenges in adhering to best practices for vaccine management, leading to increased wastage.

Routine stock monitoring of Vaccine vial monitors (VVMs) plays an important role in identifying challenges with the cold chain system and devising solutions [10,28]. Cold chain failures may expose vaccines to high temperatures if storekeepers and/or health workers do not know what to do in such cases. In developing countries like Uganda where electricity is unstable in rural area, it is common that vaccines will go to waste because of exposure to unfavourable temperature. In Cameroon, 65 health facilities lacked an alternative source of power which was significantly associated with abnormal temperature exposure [29]. Vaccine wastages attributed to high temperature exposure should be prevented through installing alternatives sources of power. A better understanding of the vaccine rates due to supply chain and cold chain failures is imperative for generating rationalisation of the UNEPI and GAVI investments in cold chain maintenance and procurement.

Some of the immunization focal persons highlighted poor reconstitution practices by new and inexperienced health workers among the reasons for vaccine wastage. Health workers' knowledge and practices of reconstitution contributes to vaccine wastage. Reconstitution practices can also lead to vaccine wastage most especially in cases where health workers are unaware of such wastages. A study assessing knowledge, attitudes and practices of vaccinators in Nigeria showed that 43% were knowledgeable about the discarding doses 6 hours after reconstitution [10]. Information gaps as result of these practices continue to increase costs associated with wastages yet can be prevented in the health system.

Health workers should be trained on how to complete vaccine monitoring tools for better data generation, quality, and utilization to support routine calculation of vaccine wastage rates. Monthly or routine collection of data on vaccine wastage rates enables health authorities to assess the efficiency of current practices, benchmark against standards, and make evidence-based decisions to improve resource allocation and policy development. Routine monitoring allows for the identification of trends over time, facilitating the evaluation of interventions and the tracking of progress in reducing wastage. Transparency and accountability are also promoted through reporting of wastage data, fostering trust within the healthcare

system. Through systematically collecting and analyzing wastage rates, authorities can identify areas for improvement and implement targeted strategies to minimize wastage, ensuring that vaccines are utilized effectively to maximize public health impact.

## Study limitations

Vaccine monitoring tools were not completed especially receipt of extra vaccines during the month from the District Vaccine Stores or other health facilities. In such instances, the number of doses vaccinated exceeded the number of available doses throughout the month at the health facility. Of note, selected health facilities were not required to have all the data available as an inclusion criterion. Overall, approximately 5% of the data pool across all facilities was excluded during data analysis due to various reasons such as missing data, extra doses and inconsistencies in reporting in the Vaccine and Material Injection Control Book (VIMCB) and tally sheets. Exclusion of this data during affected months could have led to overestimation or underestimation of vaccine wastage rates for vaccines during the affected months and overall assessment period. Furthermore, this assessment was conducted in twenty–two health facilities in two districts in the entire country due to limited funding, hence affecting generalisability of study findings to all health facilities in Uganda.

This study did not quantitatively track individual vaccine wastage events at the point of administration but rather relied on aggregated facility reports extracted from vaccine monitoring tools. Additionally, the reasons for vaccine wastage presented in the results were self-reported responses by health workers through administered questionnaires rather than from direct quantitative analysis of vaccine wastage data since these can be observed in a trial setting. These self-reported reasons may not necessarily align with the recorded wastage rates, as no validation was conducted to compare the two sources of data.

## Conclusion

This study underscores the critical challenges associated with high vaccine wastage rates, particularly for multi-dose vaccines, in Kalungu and Mukono districts of Uganda from March to August 2022. The observed variations in wastage rates among different health facilities, both in urban and rural areas, emphasize the complexity of the issue. Discarding remaining doses in opened vials of vaccines without preservatives within 6 hours of reconstitution, as per the Multi Dose Vial Policy (MDVP), emerged as a primary contributor to wastage, alongside factors such as low turnouts during vaccination outreaches and deficiencies in health worker training. The availability of vaccines in reduced-volume multi-dose vials could significantly mitigate challenges associated with high-volume multi-dose vials. Advocating for the utilization of actual or historical vaccine wastage rates, despite the possibility of surpassing acceptable ranges, while considering local practices, is vital for accurately forecasting vaccine needs and ensuring efficient resource allocation to seize every opportunity to vaccinate an eligible child. Immediate interventions, particularly in rural areas, are also necessary to strengthen the cold chain system, update micro plans, and enhance health worker knowledge, fostering a more efficient and effective immunization system in Uganda.

## Supporting information

**S1 Data. Vaccine monitoring dataset, Kalungu District, March–August 2022.**
(XLSX)

**S2 Data. Vaccine monitoring dataset, Mukono District, March–August 2022.**
(XLSX)

**S3 Data. Reasons for vaccine wastage, Kalungu District.**
(XLSX)

**S4 Data. Reasons for vaccine wastage, Mukono District.**
(XLSX)

## Author contributions

**Conceptualization:** Mackline Ninsiima, Michael Muhoozi, Henry Luzze, Simon Kasasa.

**Data curation:** Mackline Ninsiima, Michael Muhoozi, Henry Luzze, Simon Kasasa.

**Formal analysis:** Mackline Ninsiima, Michael Muhoozi, Henry Luzze, Simon Kasasa.

**Investigation:** Mackline Ninsiima, Michael Muhoozi, Henry Luzze, Simon Kasasa.

**Methodology:** Mackline Ninsiima, Michael Muhoozi, Henry Luzze, Simon Kasasa.

**Project administration:** Mackline Ninsiima, Michael Muhoozi, Henry Luzze, Simon Kasasa.

**Supervision:** Mackline Ninsiima, Michael Muhoozi, Henry Luzze, Simon Kasasa.

**Validation:** Mackline Ninsiima, Michael Muhoozi, Henry Luzze, Simon Kasasa.

**Visualization:** Mackline Ninsiima, Michael Muhoozi.

**Writing – original draft:** Mackline Ninsiima, Michael Muhoozi.

**Writing – review & editing:** Henry Luzze, Simon Kasasa.

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
