## [Decision Letter · Decision Letter 0]

8 Jul 2024

PGPH-D-24-01103

Vaccine wastage rates and attributed factors in rural and urban areas in Uganda: Case of Mukono and Kalungu districts

Dear Dr. Ninsiima,

Thank you for submitting your manuscript to PLOS Global Public Health. After careful consideration, we feel that it has merit but does not fully meet PLOS Global Public Health’s publication criteria as it currently stands. Therefore, we invite you to submit a revised version of the manuscript that addresses the points raised during the review process.

We look forward to receiving your revised manuscript.

Kind regards,

Edina Amponsah-Dacosta, Ph.D., MPH

Academic Editor

Journal Requirements:

1. Please provide additional details regarding participant consent. In the ethics statement in the Methods and online submission information, please ensure that you have specified (1) whether consent was informed and (2) what type you obtained (for instance, written or verbal, and if verbal, how it was documented and witnessed).

Additional Editor Comments (if provided):

Reviewers' comments:

Reviewer's Responses to Questions

**Comments to the Author**

1. Does this manuscript meet PLOS Global Public Health’s publication criteria?

Reviewer #1: Yes

Reviewer #2: Yes

2. Has the statistical analysis been performed appropriately and rigorously?

Reviewer #1: Yes

Reviewer #2: I don't know

3. Have the authors made all data underlying the findings in their manuscript fully available (please refer to the Data Availability Statement at the start of the manuscript PDF file)?

Reviewer #1: No

Reviewer #2: Yes

4. Is the manuscript presented in an intelligible fashion and written in standard English?

Reviewer #1: Yes

Reviewer #2: Yes

Reviewer #1: Abstract:

The mention of the evaluation period for wastage should be moved up in the earlier part of the abstract and it should clearly state that we wastage was tracked for 6 months as implied in the statement later presented that it was from March to August 2022.

For the six-month period, was this prospective tracking of wastage or retrospective. This should be made clear.

The abstract should also provide information about sample sizes for the study. Were all health facilities in the two districts included in the study?

What type of vaccine wastage was monitored and reported, open and closed vial or just open vial?

The statement in lines 30 to 31 that “Vaccine wastage rates particularly for BCG and MR vaccines were attributed to compliance with Multi Dose Vial Policy (MDVP).” is not correct as all vaccines have guidance based on this MDVP. Rather, the higher wastage rates for these vaccines is because remaining doses in opened vials of these reconstituted vaccines without preservatives have to be discarded within 6 hours of reconstitution, as per the MDVP. The same MDVP states that remaining doses in opened vials of liquid vaccines with preservatives such as IPV, can be kept for 28 days. So the issue with the high wastage rate for MR and BCG is not the adherence to the policy, but due to the nature of the vaccines.

Introduction:

In lines 79 to 81, the authors should not frame wastage as a bad thing always, given there is a tradeoff between wastage and coverage. As vaccinators try to minimize wastage, by implementing wastage reducing behaviors, such as not opening a vial for each eligible child, this also leads to missed vaccination opportunities. So the goal is not to minimize wastage as that may negatively impact coverage. I suspect national policy in Uganda is for vaccinators to open a vial even when one child is present and this increases wastage but the priority goal is increasing coverage. I would suggest that the statement you have on this on lines 85 to 87 be expanded to make it clear that wastage is not a bad thing, coverage is the priority and there is this tradeoff.

The statement on lines 90 to 91 that reads “the inflexible rule surrounding the disposal of opened

multi-dose vials at the end of a vaccination session or within a stringent six-hour timeframe….” is not correct. This is not just a stringent and unnecessary rule, this has to do with vaccine safety given the nature of the vaccines which do not have preservatives.

The assessment done in 2014, as quoted in lines 94 to 95 is 10 years old and is outdated as things could have changed in the past 10 years.

The reasons given in lines 119 to 123 seem to apply more directly to coverage rather than vaccine wastage.

Methods:

Please clarify whether this was a prospective study using new tools or was the relying on routine immunization program data?

The study seems to have relied on immunization program data. If this is correct, did all the health facilities have reporting forms? What types of wastage were tracked on the forms.

Lines 192 mentions a questionnaire, how when and how was the questionnaire completed and how frequently over the 6 month period?

Results:

Please provide some descriptive statistics of the health facilities in the sample. Information such as frequency of conducting vaccination sessions, availability of a refrigerator, catchment size etc., is helpful to contextualize the wastage rates reported.

Some of the tables and graphs don’t bring new insights and should be deleted. For example, Figure 1 duplicates a lot of the information in Table 1. The acceptable wastage rates could be added to Table 1 and this figure could be deleted. Also Figures 2 and 3 is not very informative as such trends are expected given variations in number of children vaccinated per year. Averages of the months of the evaluation are more informative rather than month to month changes. Given I don’t see the value of the trend analysis, Table 2 does not add value.

For Table 3, the results by rural urban are the same as in Table 1 given that all Kalungu is rural and all Mukono is urban and so the comparison of wastage rates for rural / urban is repeating what is stated already and should be deleted.

For Table 3, could you provide sample sizes for the subgroup based on ownership. Also provide ranges or some measure of dispersion as some of the averages need to be interpreted in the context of the small sample sizes.

Similarly for Table 4, please provide sample sizes by facility type and some measures of dispersion.

How is the p-value being calculated in Table 4, what is the comparator?

The reasons for vaccine wastage reported in the section starting on lines 314, is this based on observed wastage data or self-reported data from the health workers on what they know / think. The wastage results presented in the previous section do not disaggregate wastage rates by causes and if these data were collected, they should be reported.

Discussion:

The word remarkable on lines 404 should be deleted. This finding is not remarkable as this has been reported elsewhere and is also not surprising.

As mentioned before, the high wastage rate for BCG and MR is not due to the MDVP but rather to the vaccine presentation. Please correct lines 407 to 409. The same error is stated in line 427 and should be corrected as the application of the policy is because of vaccine safety considerations and should be adhered to! Again this incorrect information is stated on lines 435.

Lines 410, how to errors and non-completion of wastage forms result in high wastage?

The authors need to consider the tradeoff between wastage and coverage and note coverage is the higher priority. Their discussion on lines 436 to 439 need to be revised to make it clear that coverage is the priority.

I disagree with their discussion on lines 442 to 445 given the risk to missed vaccination opportunities, vials should be opened for each child present and enough vaccines should be availed for this. The policy recommendation should be to reduce the doses per vial. Please see some recent articles on this including these two:

Krudwig et al. The effects of switching from 10 to 5-dose vials of MR vaccine on vaccination coverage and wastage: A mixed-method study in Zambia. Vaccine 2020. DOI: 10.1016/j.vaccine.2020.07.012

Mvundura et al. Vaccine wastage in Ghana, Mozambique, and Pakistan: An assessment of wastage rates for four vaccines and the context, causes, drivers, and knowledge, attitudes and practices for vaccine wastage. Vaccine 2023. DOI: 10.1016/j.vaccine.2023.05.033

In line 453, single dose vials are not the solution as they have higher cold chain requirements. But rather vaccines with smaller doses per vial but still in multi dose vials.

The discussion section is long and should be streamlined to focus on the key findings and the policy implications. For example the authors discuss VVMs in lines 491 and yet their article did not mention this in any detail and so this is not part of their study. They did not characterize the availability of power at the study facilities or much else to warrant this discussion.

The allusions made in lines 509 about reconstitution are not related to wastage but rather to vaccine safety. Again, this information in these lines is incorrect as written and should be deleted.

The section on cost implications of vaccine wastage in lines 515 should also be deleted. The authors have not made such calculations in their analysis and so this should not be a discussion point. The discussion should be related to the results presented and not extrapolations from other publications.

Similarly, the discussion in lines 540 seems outside the scope of their findings and should be deleted.

Conclusion:

The wring interpretations of the meaning of the MDVP is again stated in lines 573 and should be corrected.

Some of the conclusions such as predictive modeling are not part of the study findings. This should be restricted to what is covered in the study findings.

As mentioned, single dose presentation is not the solution but rather lower vial containers. Single dose presentations present new challenges to the system and so this would be solving one problem and creating a new problem.

The authors do not include some recent wastage studies in their review. Examples of the two studies cited above are excluded and some other recent studies on the topic.

Reviewer #2: Vaccine wastage rates and attributed factors in rural and urban areas in Uganda: Case of Mukono and Kalungu districts

In this study sponsored by the East African Regional Centre of Excellence, the authors have used quantitative and qualitative data to report vaccine wastage in Uganda highlighting several known challenges with suggestions for solutions taking the local context into consideration. While some wastage is acceptable understanding the factors that contribute to wastage is key to keeping wastage to acceptable rates.

Below are some suggestions for consideration

1. Background

Some information on the schedule and vaccine vials used in country would be useful.

Which season was the data collected, rainy, dry? Give some indication

2. Methods

Study design and Sampling procedures - Consider having sub -headings for the quantitative and qualitative component.

Consider the same for data collection and data analysis

Besides one being rural and the other urban it is not clear how the two districts fit in the rest of the county. A map will help to show how these are placed in the rest of the country. Were these also chosen purposefully?

3. Results

Line 314- 320 Reason for vaccine wastage

Were these results from the interviews with the health workers? . In that case, it would be better to move it to the section after the characteristics of the key informants. If not please clarify in the methods how these data were collected.

Why do you having the footnote in table 4 with the p values, these can be seen in the table.

4. Discussions

Line 453 – 455 - Though the challenge of cost was mentioned elsewhere in the manuscript and here consideration on feasibility of single dose vials is mentioned, perhaps a statement on the associated costs of single vial dose - increased storage space, higher vaccine cost could additionally be highlighted in the document.

5. References

Review the references. There are some duplicates

Reference 5 and 23,

Reference 22 and 28 are the same

Reference 7 is inappropriate for the statement in the text.

**Do you want your identity to be public for this peer review?** For information about this choice, including consent withdrawal, please see our Privacy Policy

Reviewer #1: No

Reviewer #2: No

---

## [Decision Letter · Decision Letter 1]

9 Feb 2025

PGPH-D-24-01103R1

Vaccine wastage rates and attributed factors in rural and urban areas in Uganda: Case of Mukono and Kalungu districts

Dear Dr. Ninsiima,

Thank you for submitting your manuscript to PLOS Global Public Health. After careful consideration, we feel that it has merit but does not fully meet PLOS Global Public Health’s publication criteria as it currently stands. Therefore, we invite you to submit a revised version of the manuscript that addresses the points raised during the review process.

The manuscript has been evaluated by two reviewers, and their comments are available below.

The reviewers have raised a number of major concerns. They request improvements to the reporting of methodological aspects of the study, particularly regarding the prospective data collection for vaccine wastage, as well as improvements to the writing and discussion.

Could you please carefully revise the manuscript to address all comments raised?

We look forward to receiving your revised manuscript.

Kind regards,

Helen Howard

Staff Editor

Journal Requirements:

1. Please provide additional details regarding participant consent. In the ethics statement in the Methods and online submission information, please ensure that you have specified (1) whether consent was informed and (2) what type you obtained (for instance, written or verbal, and if verbal, how it was documented and witnessed).

Additional Editor Comments (if provided):

Reviewers' comments:

Reviewer's Responses to Questions

**Comments to the Author**

Reviewer #1: (No Response)

Reviewer #2: All comments have been addressed

publication criteria?

Reviewer #1: Yes

Reviewer #2: Yes

3. Has the statistical analysis been performed appropriately and rigorously?

Reviewer #1: Yes

Reviewer #2: I don't know

4. Have the authors made all data underlying the findings in their manuscript fully available (please refer to the Data Availability Statement at the start of the manuscript PDF file)?

Reviewer #1: Yes

Reviewer #2: Yes

5. Is the manuscript presented in an intelligible fashion and written in standard English?

Reviewer #1: Yes

Reviewer #2: Yes

Reviewer #1: The authors have done a great job addressing the majority of the comments provided. However, a few prior comments were not adequately addressed.

The methodology for the prospective data collection for vaccine wastage is still not clear. As we know, monitoring of vaccine usage / wastage is poor. So, can the authors better describe the process used by the study team during the six-month period to ensure that the health workers were monitoring and documenting vaccine usage? Was anything done to remind health workers to monitor and document vaccine usage? Or did the study team rely on routine data which that collected as usual (though we know monitoring of vaccine usage and wastage is poor due to several reasons such as lack of training, fear of high wastage rates as a negative performance indicator etc.). What was different at the study health facilities in the six-month study period versus what they routinely do? How can we know that the data over the six-month period reflect true vaccine usage and then wastage rates?

The response the authors provided in the response document to my question on whether the reasons for vaccine wastage presented in the results section are from observed wastage data or self-report should be explicitly stated in the revised manuscript. You mention in the response that “The results referred to were obtained from administering questionnaires to health workers providing vaccination services in the selected health facilities”. And so please state this in the revised manuscript otherwise readers can think these reasons are from the quantitative data on wastage rates and yet these are more the opinions of the health workers, which may or may not align with the quantitative data as this has not been validated.

Please add to the limitations that you did not quantitative track wastage rate, as per your response to one of my prior questions and also to clarify the point above that the reasons for wastage are not from quantitative wastage data but self-report from health workers / vaccinators.

Reviewer #2: Vaccine wastage rates and attributed factors in rural and urban areas in Uganda: Case of Mukono and Kalungu districts

The authors have provided estimates of vaccine wastage from two districts in Uganda and used qualitative approach to provide factors associated with vaccine wastage. The factors reported are not new, however the paper highlights local challenges for these districts in Uganda.

Here are some comments for consideration

1. In general there are several repetitions in the manuscript for example, Line 48/49 ‘in Uganda’ repeated, see ….line 168/69 for another repetition. They authors should review the manuscript and remove unnecessary duplication.

2. A map showing the selected districts and or major health facilities would be helpful

3. Page 31 and 32. Some references are duplicated for example ref 5 and xxx. This looks like an error in the revised manuscript. Please review

4. The qualitative study confirmed low turnout for vaccination during the rainy season. Are six months within the rainy season in Uganda do they fall within the rainy season? This will affect the trends. Please include information on rainy season in relation to the months when the study was conducted.

5. It appears that for resource constraints two districts were selected purposefully. However, it would be good to have some justification for the sample size. Why did you choose a third of the facilities in each district, was this based-on any proportion/size? How did you decide on the total numbers in the different categories to be chosen?

6. Are Mobile vaccination clinics common in Uganda?

7. Line 417, what is the point of referring to recommendation by Min of Health in India?

8. Line 435/6 give more context to the comparison of IPC and OPV vaccine vial presentation in the setting you are referring to.

9. There are several statements without supporting references/ in appropriate ref

Line 70-72 provide references

Line 75 ref 6 is inappropriate

Line 79 provide a ref

Line 119-125 provide some references

Line 436-38 provide some references

**Do you want your identity to be public for this peer review?** For information about this choice, including consent withdrawal, please see our Privacy Policy

Reviewer #1: No

Reviewer #2: No

---

## [Decision Letter · Decision Letter 2]

30 Apr 2025

Vaccine wastage rates and attributed factors in rural and urban areas in Uganda: Case of Mukono and Kalungu districts

PGPH-D-24-01103R2

Dear Ms. Ninsiima,

We are pleased to inform you that your manuscript 'Vaccine wastage rates and attributed factors in rural and urban areas in Uganda: Case of Mukono and Kalungu districts' has been provisionally accepted for publication in PLOS Global Public Health.

Best regards,

Julia Robinson

Executive Editor

Reviewer Comments (if any, and for reference):

Reviewer's Responses to Questions

**Comments to the Author**

Reviewer #1: All comments have been addressed

Reviewer #2: All comments have been addressed

publication criteria?

Reviewer #1: Yes

Reviewer #2: Yes

3. Has the statistical analysis been performed appropriately and rigorously?

Reviewer #1: Yes

Reviewer #2: I don't know

4. Have the authors made all data underlying the findings in their manuscript fully available (please refer to the Data Availability Statement at the start of the manuscript PDF file)?

Reviewer #1: Yes

Reviewer #2: Yes

5. Is the manuscript presented in an intelligible fashion and written in standard English?

Reviewer #1: Yes

Reviewer #2: Yes

Reviewer #1: Thanks for addressing all the comments.

Reviewer #2: (No Response)

**Do you want your identity to be public for this peer review?** For information about this choice, including consent withdrawal, please see our Privacy Policy

Reviewer #1: No

Reviewer #2: No
